# Gut Microbiota Bacterial Species Associated with Mediterranean Diet-Related Food Groups in a Northern Spanish Population

**DOI:** 10.3390/nu13020636

**Published:** 2021-02-16

**Authors:** Carles Rosés, Amanda Cuevas-Sierra, Salvador Quintana, José I. Riezu-Boj, J. Alfredo Martínez, Fermín I. Milagro, Anna Barceló

**Affiliations:** 1Servei de Genòmica i Bioinformàtica, Universitat Autònoma de Barcelona, 08193 Bellaterra, Spain; carles.roses@uab.cat; 2Center for Nutrition Research, Department of Nutrition, Food Sciences and Physiology, University of Navarra, 31008 Pamplona, Spain; acuevas.1@alumni.unav.es (A.C.-S.); jiriezu@unav.es (J.I.R.-B.); jalfmtz@unav.es (J.A.M.); fmilagro@unav.es (F.I.M.); 3Independent Researcher, 08021 Barcelona, Spain; squintana@mutuaterrassa.es; 4Navarra Institute for Health Research (IdISNA), 31008 Pamplona, Spain; 5Centro de Investigación Biomédica en Red de la Fisiopatología de la Obesidad y Nutrición (CIBERobn), Instituto de Salud Carlos III, 28029 Madrid, Spain

**Keywords:** *Bifidobacterium animalis*, gut microbiota, short-chain fatty acids, obesity, butyrate

## Abstract

The MD (Mediterranean diet) is recognized as one of the healthiest diets worldwide and is associated with the prevention of cardiovascular and metabolic diseases. Dietary habits are considered one of the strongest modulators of gut microbiota, which seem to play a significant role in health status of the host. The purpose of the present study was to evaluate interactive associations between gut microbiota composition and habitual dietary intake in 360 Spanish adults from the Obekit cohort (normal weight, overweight, and obese participants). Dietary intake and adherence to the MD tests were administered and fecal samples were collected from each participant. Fecal 16S rRNA (ribosomal Ribonucleic Acid) gene sequencing was performed and checked against the dietary habits. MetagenomeSeq was the statistical tool applied to analyze data at the species taxonomic level. Results from this study identified several beneficial bacteria that were more abundant in the individuals with higher adherence to the MD. *Bifidobacterium animalis* was the species with the strongest association with the MD. Some SCFA (Short Chain Fatty Acids) -producing bacteria were also associated with MD. In conclusion, this study showed that MD, fiber, legumes, vegetable, fruit, and nut intake are associated with an increase in butyrate-producing taxa such as *Roseburia faecis, Ruminococcus bromii,* and *Oscillospira (Flavonifractor) plautii.*

## 1. Introduction

Gut microbiota status has an impact on the health and disease of the host [1]. Dietary habits are considered one of the strongest modulators of gut microbiota. However, it is not clear how the two relate to each other or which gut microbiota profile to pursue in order to ensure good health. Currently, the majority of the world’s population is exposed to Western diets, characterized by a high intake of saturated and omega-6 fatty acids, reduced omega-3 and fiber intake, an overuse of salt, and too much refined sugar and processed food [2]. Along with a sedentary lifestyle, these factors are increasing the prevalence of obesity worldwide, with half of the world’s population now considered to be overweight [3]. As a result of this lifestyle, serious conditions can appear and in a variety of ways. Hypertrophied adipocytes release inflammatory molecules (i.e., interleukins and tumor necrosis factor), which can act as false alarms in the immune system, causing the entire immune system to reduce its sensitivity, such that the response to a real condition may be delayed [4]. This can aid the development of several inflammation-related disorders such as metabolic syndrome, cardiovascular disease, colorectal cancer, and neurodegenerative diseases [5,6]. Most of these disorders have also been associated with alterations in microbiota composition in humans, especially those with reduced bacterial richness and diversity [7]. These changes have been related to disturbed gut barrier functions, increased gut permeability, and increased plasma concentrations of lipopolysaccharide (LPS) and other bacterial by-products, which cause low-grade inflammation that, again, triggers the development of insulin resistance, obesity, metabolic syndrome, colorectal cancer [8], and autoimmune disorders such as Crohn disease, ulcerative colitis, and allergies [7].

In this context, the MD (Mediterranean diet) is recognized as one of the healthiest diets worldwide as it contains to a high proportion of fiber, mono- and poly-unsaturated fatty acids, antioxidants, and polyphenols, present in vegetables, fruits, pulses, and extra virgin olive oil (EVOO), which are all strongly associated with a reduced risk of developing non-communicable diseases related to Western diet and lifestyle [9,10]. In a study with overweight, obese participants with lifestyle risk factors for metabolic disease, an isocaloric MD intervention reduced their blood cholesterol and caused multiple changes in their microbiome and metabolome, thus improving their metabolomic health [11]. A 12-month-long MD intervention with elderly subjects showed a taxa enrichment that was associated with lower frailty and improved cognitive functions, but negatively correlated with inflammatory markers, thus promoting healthier aging [12]. Carbohydrates and fiber present in the MD are fermented by gut microbiota, through which large quantities of biologically active metabolites such as SCFAs (short-chain fatty acids) are produced [13]. Also, greater concentrations of phenolic metabolites are excreted in feces when there are high concentrations of bioactive compounds coming from polyphenols and fiber [14]. Furthermore, de Filippis et al. reported that a high adherence to a MD rich in plant foods beneficially impacts gut microbiota and the associated metabolome [15]. Therefore, we would expect a modulation of the gut microbiota to be one of the positive health effects of the MD [13]. Mitsou et al. found a positive correlation with gastrointestinal symptoms, fecal moisture, total bacteria, and Bifidobacteria, but a reduced representation of Lactobacilli and butyrate-producing bacteria induced by fast food consumption [16].

The main objective of the present study was to relate the adherence to the MD to specific metagenomic traits, focusing on those bacterial taxa that are more abundant in individuals with a high adherence to the MD. In addition, a specific food group consumption assessment was carried out to increase our knowledge of the impact of diet on gut microbiota composition. The focus was on the bacteria that are most closely associated with a high adherence to the MD and how these bacteria are influenced by specific food groups characteristic of the Mediterranean pattern.

## 2. Material and Methods

### 2.1. Participants

This cross-sectional study enrolled 360 Spanish adults of self-reported European ancestry (251 females and 109 males) with ages ranging from 45.0 ± 10.5 years old. Participants were recruited at the Center for Nutrition Research of the University of Navarra, Spain, and took part in the Obekit study. Major exclusion criteria included a history of diabetes mellitus, cardiovascular disease and hypertension, pregnant or lactating women, and current use of lipid-lowering drugs. Patients with a diagnosis of primary hyperlipidemia were also excluded. This investigation followed the ethical principles for medical research in humans from the 2013 Helsinki Declaration [17]. The research protocol (ref. 132/2015) was approved by the Research Ethics Committee of the University of Navarra. Written informed consent from each participant was obtained before the inclusion in the study. The characteristics of the Obekit research project, including study design and registration, have been reported elsewhere [18].

Inclusion criteria were body mass index between 25 and 40, a physical examination and assessment of vital signs considered as normal or clinically insignificant by the researcher. Also, in the case of individuals with chronic, stable-dose drug treatment during the last three previous months and at baseline, the investigators assessed possible inclusion.

### 2.2. Anthropometric and Biochemical Measurements

Anthropometric measurements such as height (cm) and body weight (kg) were collected in the fasting state by trained nutritionists following validated procedures [19]. Body mass index (BMI) was calculated as the ratio between weight and squared height (kg/m^2^), and, according to the WHO (World Health Organization) standards, the volunteers were classified as normal weight when BMI was 18.5–24.9 (*n* = 64), overweight when BMI was 25.0–29.9 (*n* = 115), and obese when BMI > 30.0 (*n* = 181).

Venous blood samples were drawn by venipuncture in a clinical setting after an overnight fast. Blood biochemistry (glucose, total cholesterol (TC), high-density lipoprotein cholesterol (HDL), and triglycerides) was analyzed with a Pentra C200 clinical chemistry analyzer (HORIBA Medical, Madrid, Spain) and suitable kits provided by the company. Low-density lipoprotein cholesterol (LDL-c) was estimated using the Friedewald equation: LDL-c = TC-HDL-(triglycerides/5). Insulin was measured using a specific enzyme-linked immunosorbent assay (Mercodia, Uppsala, Sweden) read with an automated analyzer system (Triturus, Grifols, Barcelona, Spain). The homeostatic model assessment for insulin resistance (HOMA-IR) was calculated as follows: fasting insulin (microU/L) × fasting glucose (nmol/L)/22.5.

### 2.3. Dietary Estimation

Habitual dietary intake at baseline was collected with a validated food frequency questionnaire that included 137 food items with corresponding portion sizes [20]. All participants enrolled were asked to provide information about the number of times they had consumed each food item during the previous year according to four frequency categories: daily, weekly, monthly, or never. Total energy (kcal) and macronutrient intakes (%) were determined with ad hoc software and the information available from valid Spanish food composition tables [21]. Among all this information, six groups were defined as the most representative of the MD: fiber, legumes, vegetables, fruit, olive oil, and nuts. Consumption of these groups was represented in g per day and these scores were transformed into tertiles as cut-off points. The second tertile was removed and first and third were compared. A 14-item questionnaire, PREDIMED (Prevención con Dieta Mediterránea) validated test, was also used in this study to appraise the adherence of participants to the MD [22]. The MD score ranged from 0 (minimal adherence) to 14 (maximal adherence). Ranges of scores were defined again by tertiles: first tertile (0–6 score) for low, second tertile (7–8) was removed, and third tertile (9–14) for high adhesion to the MD.

### 2.4. Faecal Sample Collection and DNA Extraction

Volunteers self-collected fecal samples at baseline using OMNIgene.GUT kits from DNA Genotek (Ottawa, ON, Canada), according to the standard instructions provided by the company. The isolation of DNA from fecal samples was performed with QIAamp^®^ DNA kit (Qiagen, Hilden, Germany), following the manufacturer’s protocol.

### 2.5. Metagenomic Data: Library Preparation

Bacterial DNA sequencing was performed by the Servei de Genòmica i Bioinformàtica from the Universitat Autònoma de Barcelona (Bellaterra, Cerdanyola del Vallés, Spain). Metagenomics studies were performed by analyzing the variable regions V3–V4 of the prokaryotic 16S rRNA (ribosomal Ribonucleic Acid) gene sequences, which gives 460 bp amplicons in a two-round PCR protocol.

In a first step, PCR is used to amplify a template out of a DNA sample using specific primers with overhang adapters attached that flank regions of interest. The full-length primer sequences, using standard IUPAC (International Union of Pure and Applied Chemistry) nucleotide codes, to follow the protocol targeting this region were: **Forward Primer**: 5′TCGTCGGCAGCGTCAGATGTGTATAAGAGACAGCCTACGGGNGGCWGCAG and **Reverse Primer:** 5′GTCTCGTGGGCTCGGAGATGTGTATAAGAGACAGGACTACHVGGGTATCTAATCC. PCR was performed in a thermal cycler using the following conditions: 95 °C for 3 min, 25 cycles of (95 °C for 30 s, 55 °C for 30 s, and 72 °C for 30 s), and 72 °C for 5 min.

To verify that the specific primers had been correctly attached to the samples, 1 µL of the PCR product was checked on a Bioanalyzer DNA 1000 chip (Agilent Technologies, Santa Clara, CA, USA). The expected size on a Bioanalyzer is ~550 bp.

In a second step and using a limited-cycle PCR, sequencing adapters, and dual indices barcodes, Nextera^®^ XT DNA Index Kit, FC-131-1002 (Illumina, San Diego, CA, USA), were added to the amplicon, which allows up to 96 libraries for sequencing on the MiSeq sequencer with the MiSeq^®^ Reagent Kit v3 (600 cycle) MS-102-3003 to be pooled together.

PCR was performed in a thermal cycler using the following conditions: 95 °C for 3 min, eight cycles of (95 °C for 30 s, 55 °C for 30 s, and 72 °C for 30 s), and 72 °C for 5 min. Subsequently, the Index PCR ran a second Bioanalyzer DNA 1000 chip to validate the library. The expected size was ~630 bp.

The next step consisted of the quantification of the libraries using a fluorometric quantification and dilution of the samples before pooling all samples.

Finally, paired-end sequencing was performed on a MiSeq platform (Illumina) with a 600 cycles Miseq run [23] and with 20 pM sample and a minimum of 20% PhiX. The mean reads obtained were 164,387. Only samples with more than 40,000 reads were used for further analysis. All the sequencing data were deposited by the authors in SRA (Sequence Read Archive) and the accession key has been included in the text (PRJNA623853).

### 2.6. Metagenomics Data: Analysis and Processing

The 16S rRNA gene sequences obtained were filtered following the quality criteria of the OTUs (operational taxonomic units) processing pipeline LotuS (release 1.58) [24]. This pipeline includes UPARSE (Highly accurate OTU sequences from microbial amplicon reads) de novo sequence clustering and removal of chimeric sequences and phix contaminants for the identification of OTUs and their abundance matrix generation [25,26]. Taxonomy was assigned using HITdb (Highly scalable Relational Database), achieving up to species sensitivity. BLAST (Basic Local Alignment Search Tool) was used when HITdb failed to reach a homology higher than 97% [27,28]. Thus, OTUs with a similarity of 97% or more were referred to species. However, OTUs that did not reach this percentage of similarity were checked and updated using the Basic Local Alignment Search Tool (BLASTn) to compare with the 16S rRNA gene sequences for bacteria and archaea database of GenBank of National Center for Biotechnology information in order to find an assignment to a species. These sequences in which the BLASTn tool found a new assignment were indicated using the GenBank access number and the percentage of homology following the species name. The abundance matrices were first filtered and then normalized in R/Bioconductor at each classification level: OTU, species, genus, family, order, class, and phylum. This study focused mainly on the species level. Briefly, taxa with a less than 10% frequency in our population were removed from the analysis, and a global normalization was performed using the library size as a correcting factor and log2 data transformation [29].

### 2.7. Richness and Evenness

Richness was defined as the total of species. Evenness was calculated using Pielou’s evenness index according to the following formula: J’ = H/ln(S).

Alpha diversity was assessed using the Shannon index. Beta diversity was calculated using the Bray–Curtis index, PERMANOVA (Permutational Multivariate Analysis of Variance) statistical method, and NMDS (Non-metric Multidimensional Scaling) as ordination methods.

All the sequencing data were deposited by the authors in the Sequence Read Archive (SRA) from NCBI (National Center for Biotechnology Information), with PRJNA623853 as accession key.

### 2.8. Statistical Analysis

The microbiome Analyst tool [30] was used for statistical differences in microbiota profiles between groups (tertiles) through a Zero-inflated Gaussian approach of MetagenomeSeq and using the cumulative sum scaling (CSS) normalization.

### 2.9. Prediction of Functional Potential of Gut Microbiota

Computational prediction of the functional capabilities using data from 16S rRNA metagenomics was performed using the Tax4fun tool from MicrobiomeAnalyst [30]. KEGG (Kyoto Encyclopedia of Genes and Genomes) Orthologus (KO) provided by Tax4Fun was comparatively analyzed using the Shotgun Data Profiling section in the MicrobiomeAnalyst. A total of 2094 KO low-abundance features were removed based on prevalence (<20% of prevalence in samples) and 336 low-variance features based on inter-quantile range, with a total of 3018 features remaining after filtering. The EdgeR statistical analysis compared differentially expressed KO between groups (low and high adherence to MD) and allowed to visualization of the results within KEGG metabolic networks, along with pathway analysis between groups.

## 3. Results

### 3.1. Participant Characteristics

Baseline characteristics of the population that participated in this study separated by adherence to the MD and BMI (normal weight, overweight, and obese, according to the World Health Organization criteria [31]) are shown in Table 1, including age, anthropometric measures, and biochemical and dietary data. Additional data on the study population (separated by sex and adiposity status) are shown as Appendix A.

The high-adherence group was made up of older people, with a significantly lower BMI and less resistance to insulin. Despite the total energy intake being very close, the data in Table 1 indicate that the third tertile values were healthier. Concerning dietary composition, the third tertile was characterized by a higher intake of fiber.

### 3.2. Microbiota Composition: MD Adherence

MD tertiles 1 and 3 were compared through metagenomeSeq analysis. Significant differences appeared when comparing both tertiles (FDR (False Discovery Rate) < 0.05). Species shown in Table 2 were strongly influenced by the MD score. Participants with a higher adherence to the MD were represented in the third tertile while those who are far from the MD model were in the first tertile.

This work focused on the high-adherence species and their distribution, with box plots (Figure 1). All box plots represent those species with significant differences between high- and low-adherence tertiles to the respective dietary pattern or food group.

According to the species shown in Table 2, the relationship between these bacteria and the intake of certain foods (g of food per day) was analyzed. The food groups chosen for this approach were those most relevant in the MD: legumes, vegetables, fruit, olive oil, nuts, and also total fiber.

### 3.3. Microbiota Composition: Fibre Intake

It has been proposed that a substantial part of the beneficial effects of the MD could be attributed to a high intake of fiber. Species upregulated in the high MD adherence group were analyzed to see which of them were related to fiber intake (FDR < 0.05) (tertiles 1 and 3). The most abundant species in the tertile of higher fiber intake (third tertile) are shown in Table 3 and Figure 2.

### 3.4. Microbiota Composition: Food Groups

Focusing on the species influenced by the MD (Table 2), those positively influenced by legumes, vegetables, fruit, and nut intake (FDR < 0.05) are shown in Table 3 and Figure 3, Figure 4, Figure 5 and Figure 6, respectively. No species from the high-adherence MD group were found in relation to the intake of olive oil.

### 3.5. Species Richness, Evenness, and Diversity

The total number of species obtained in this study was 4733. Richness, calculated as the number OTUs for each individual, was 535 ± 17.25 for the high and 536 ± 16.84 for the low group (*p =* 0.960). Evenness, as calculated with the Pielou index, was 0.315 ± 0.007 for the high- and 0.303 ± 0.007 for the low-adherence group (*p =* 0.220). Alpha diversity was assessed using the Shannon index, but no significant differences were obtained when comparing low- and high-adherence groups to the MD (*p =* 0.220). As for beta diversity, no significant differences were observed when comparing low-adherence and high-adherence groups to the MD (*p =* 0.554). The NMDS (Non-metric multidimensional scaling) graph is shown as Appendix A.

### 3.6. Comparison of Functional Potential of the Gut Microbiota

The analysis by Tax4Fun of differential KO abundance between low-adherence and high-adherence groups to the MD revealed a total of five KO (shown in Table 4) that were significantly different (FDR < 0.05). Interestingly, one of these (K02106) was a short-chain fatty acid transporter. The enrichment analysis showed that the sphingolipid metabolism pathway (ko00600), associated with K00720, was significantly upregulated in the high-adherence group (*p* = 0.01).

## 4. Discussion

It is widely known that the gut microbiota co-develops with the host, and its bacterial proportions are modified by the action of diet and other extrinsic stressors [32], thus determining gut microbiota composition, diversity, and activity [33]. However, despite the fact that diet and health are interrelated, there is little information on the impact of specific components of the diet on microbiota composition. The present study focused on the effect of the MD and its most characteristic food groups (fiber, legumes, vegetables, fruit, and nuts) on gut microbiota composition.

## 5. MD High-Adherence Species

There is a large amount of written evidence that shows a high adherence to the MD to be beneficial to human health. The MD is a great resource in helping to manage obesity-related comorbidities, such as cardiovascular diseases, type 2 diabetes, and pro-inflammatory conditions [34,35,36]. Table 2 shows those species that are more strongly associated with the adherence to the MD. The high-adherence group represents those participants that have a diet closer to the MD model. The following species are the most significant.

*Bifidobacterium animalis* belongs to the phylum Bacteroides. Existing data reveal that this phylum has been associated with obesity-related abnormalities in bacterial gut microbiota and that the genus *Bifidobacterium* might play a critical role in weight regulation [37]. *B. animalis* subsp. *lactis* GCL2508 is a probiotic strain capable of proliferating and producing SCFA in the gut. SCFA are the result of the fermentation of non-digestive polysaccharides in the colon, thanks to bacteria. A study with mice showed increased levels of SCFA were present as a result of treatment with this probiotic, enhancing host energy expenditure and decreasing fat accumulation [38]. These compounds have a regulatory effect on inflammatory conditions [39]. Furthermore, *B. animalis* subsp. *lactis* may have an anti-metabolic syndrome effect [38]. In our study, a K0 (K02106) defined as a SCFA transporter was overrepresented in the group with higher adherence to the MD.

*Bacteroides cellulosilyticus* received its name because of its ability to degrade cellulose [40]. It is equipped with an unprecedented number of carbohydrate-active enzymes, more than any previously sequenced members of the Bacteroidetes, providing a versatile carbohydrate utilization with a strong emphasis on plant-derived xylans abundant in cereal grains [41]. In our study it was positively correlated with legume intake (Table 3).

*Paraprevotella clara,* a common member of the human intestinal microbiota [42], is closely related to the carbohydrate-active enzymes of arabinofuranosidase, pectin lyase, and polygalacturonaseand xylanase [43] known to degrade insoluble fiber [44]. Indeed, *P. clara* is known to produce acetic acid [42]. No significant relationship with fiber was found in our results.

*Oscillibacter valericigenes* produces valeric acid, an SCFA [45]. Valeric acid has been reported to have an inhibitory effect on histone deacetylase (HDAC) isoforms implicated in a variety of disorders such as cancer, colitis, and cardiovascular and neurodegenerative diseases [46]. In this study, *O. valericigenes* showed a positive relationship with fiber consumption (Table 3). Another study observed a significant increase of this species in participants fed resistant starch-rich diets [47].

High levels of *Oscillospira (Flavonifractor) plautii* have been strongly correlated with a high production of SCFA, especially propionate and butyrate [48]. This species is of particular interest since its abundance is found to correlate with a lean host phenotype [49]. Furthermore, the *Oscillospira* genus has been seen to correlate with the production of secondary bile acids known to prevent *Clostridium difficile*-associated infectious diseases in humans [50]. It appears in Table 3 and is positively correlated with fiber intake.

*Roseburia faecis* is a butyrate producer whose abundance has been related to weight loss and a reduced glucose intolerance in mice [51]. This OTU shows a positive correlation with fiber, fruit, and nut intake (Table 3).

*Catabacter hongkongensis,* commonly found in the human intestinal microbiota [42,52], is positively influenced by fiber, legumes, vegetables, and fruit intake (Table 3).

*Ruminococcus bromii* has been linked to diets rich in fiber and resistant starch and greatly contributes to butyrate production in the colon [53]. It is positively correlated with legumes’ intake (Table 3), which are rich in resistant starch. It is important to highlight some beneficial effects of butyric acid as it has been reported to improve the intestinal barrier integrity [54], regulate cell apoptosis [55], stimulate production of anaerobic hormones [56], and, by inducing differentiation of colonic regulatory T cells, suppress inflammatory and allergic responses [57]. Furthermore, many conditions have been associated with low levels of butyrate, such as colon cancer or obesity [53]. Thus, increased butyrate production in the colon may be beneficial to human health.

*Erysipelatoclostridium ramosum* is a member of the Erysipelotrichaceae family known to interfere in various ways with the enterohepatic circulation and excretion of bilirubin, transforming it into urobilin [58]. Although this species has previously been linked to increased fatty acid absorption [59,60] and systemic inflammation [61], this sequence also shares a close homology with other species such as *Erysipelatoclostridium saccharogumia* and *Clostridium cocleatum*. *E. saccharogumia* is a lignin-converting bacterium related to anticancer [62] and osteoprotective effects [63], whereas *C. cocleatum* plays a role in mucin degradation and shows resistance to colonization by *Clostridium difficile* [64].

*Papillibacter cinnamivorans* is not well known but has been found to be more frequently present, in lower amounts, in centenarians than in any other age groups. In our study, it was shown to have a positive correlation with vegetable and nut intake (Table 3) [65].

A strong adherence to the MD and a high intake of the food groups analyzed in this study clearly modulated the microbiota population profile toward a healthier one. Although several studies have also analyzed the effect of healthy diets, such as the Nordic [66] or the Japanese [67] diets, on gut microbiota composition, it is difficult to compare the results, as the methodologies used were different and the results may not be extrapolated from one study to another due to different classifications: enterotypes, other taxonomic levels, pathogenic bacteria, specific biomarkers, etc. In any case, the present results provide evidence for the notion that a high adherence to the MD leads to increased levels of several fiber- and carbohydrate-degrading bacterial species linked to SCFA metabolism (especially butyrate, which has an anti-inflammatory effect) that may contribute toward a healthier status. Several studies support these results [10,11,12,16].

One of the reasons for the novelty of the present results may be the specificity of the study population, who were all from a Northern Spanish region and had a specific dietary pattern (close to the MD) that may condition their microbiota profile. In any case, our study revealed that several species that are involved in the production of SCFA, such as *Ruminococcus bromii*, *Roseburia faecis*, *Oscillospira (Flavonifractor) plautii*, *Oscillibacter valericigenes*, *Paraprevotella clara*, and especially *Bifidobacterium animalis*, are more abundant in the Spanish population with a higher adherence to the MD. Furthermore, we were able to associate some of these with the intake of specific food groups. However, we did not observe genera, families, or phyla that significantly differed (FDR < 0.05) between the lowest and highest tertiles of the MD.

However, it remains unclear whether a higher amount of some species is always linked to a healthier status. It seems necessary to perform a more detailed taxonomic classification, as a better classification of OTUs into subspecies would help us to understand why certain species seem to be positively influenced by the MD but might not have positive effects on human health. Also, further research is needed to understand how the growth of specific bacteria can affect microbiota composition and function. Does growth of a beneficial bacterium always mean better health? When is the balance lost toward dysbiosis? These are some of the unknowns for future studies to elucidate.

## 6. Study Limitations

Following the inclusion criteria of the project, participants included in this study were all overweight or obese, with similar related comorbidities. Other factors such age, socioeconomic conditions, educational, and psychosocial factors were also similar. All participants were from Northern Spain, which may potentially introduce bias in relation to their dietary habits, influenced by the gastronomic culture and habits of that region. Moreover, a larger sample size including participants from other Mediterranean countries could be considered for further studies. The analysis of the V3–V4 regions of the 16S rRNA gene sequence is commonly used in studies aimed at identifying bacterial genera and species, but there are other techniques, such as Nanopore sequencing (Oxford Nanopore^®^) or SMART sequencing (PACBIO—Pacific Biosciences^®^), that allow sequencing the whole 16s rRNA gene, which, in some cases, leads to subspecies’ identification. Furthermore, a full metagenomic analysis is possible using Shotgun Metagenome Analysis. Finally, we were not able to quantify SCFA in the plasma samples since, when they were extracted and stored, this objective had not been established at that time.

## 7. Conclusions

Our results indicate that the well-known beneficial factors of a MD may be triggered by changes in intestinal microbiota due to dietary habits. A high adherence to the MD seems to increase the abundance of some species associated with good health. *Bifidobacterium animalis* is the species with the strongest association with the MD. Fiber intake enhances the growth of several SCFA-producing species, such as *Oscillibacter valericigenes*, *Oscillospira (Flavonifractor) plautii,* and *Roseburia faecis. R. faecis* is also enhanced by fruit and nut consumption. Legumes enhance *Ruminococcus bromii* and vegetables increase the *Butyricicoccus pullicaecorum* population. Nut intake benefits *Papillibacter cinnamivorans* growth. This study strongly suggests that a MD and an intake of fiber, legumes, vegetables, and fruit increase butyrate production from *R. faecis, R. bromii,* and *Oscillospira (Flavonifractor) plautii. Erysipelatoclostridium ramosum* is the only bacteria from this study that does not show a clear beneficial effect on health, although this finding should be interpreted with caution. A more detailed taxonomy is required in order to come to definitive conclusions.

## Figures and Tables

**Figure 1 nutrients-13-00636-f001:**
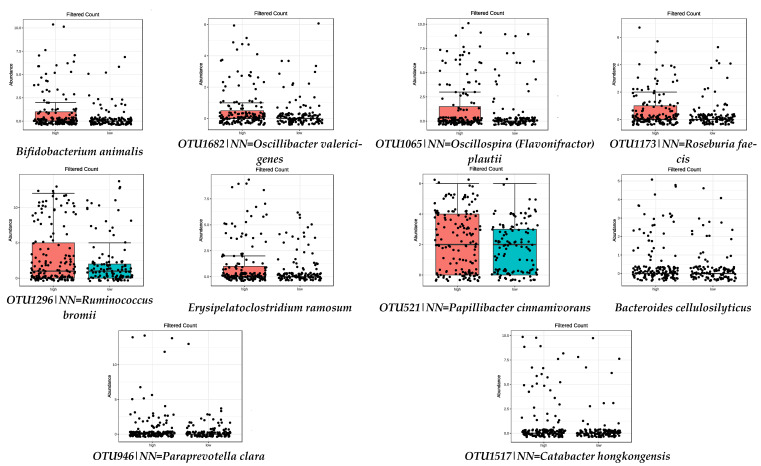
Bacterial species that were significantly more abundant in the group with high adherence to MD (FDR < 0.05) by metagenomeSeq test. Red boxes represent participants with a higher adherence to the MD and blue boxes low adherence.

**Figure 2 nutrients-13-00636-f002:**
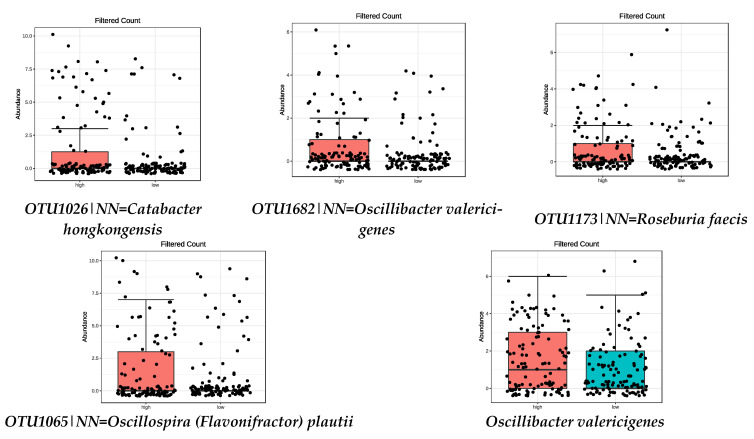
Bacterial species that show significant relation with high FIBRE intake (FDR < 0.05) by metagenomeSeq test. Red boxes represent participants with a higher adherence to the MD and blue boxes low adherence.

**Figure 3 nutrients-13-00636-f003:**
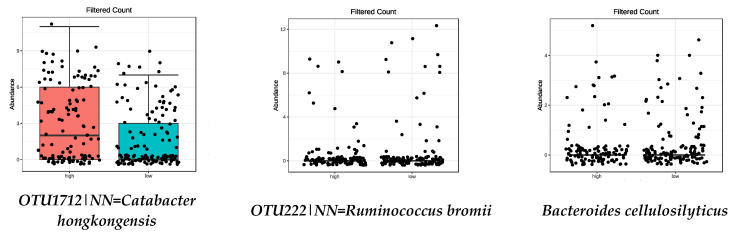
Bacterial species that were significantly more abundant in the group with high LEGUMES intake (FDR < 0.05) by metagenomeSeq test. Red boxes represent participants with a higher adherence to the MD and blue boxes low adherence.

**Figure 4 nutrients-13-00636-f004:**
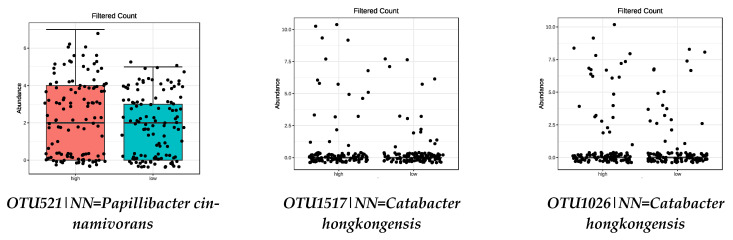
Bacterial species that were significantly more abundant in the group with high VEGETABLES intake (FDR < 0.05) by metagenomeSeq test. Red boxes represent participants with a higher adherence to the MD and blue boxes low adherence.

**Figure 5 nutrients-13-00636-f005:**
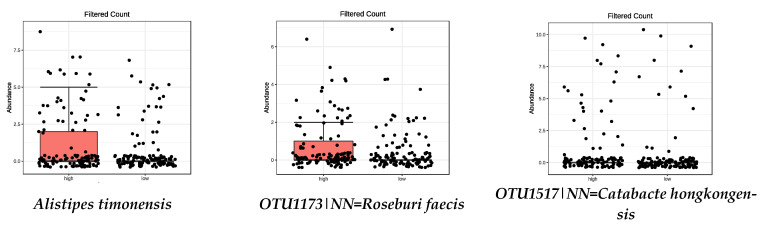
Bacterial species that were significantly more abundant in the group with high FRUIT intake (FDR < 0.05) by metagenomeSeq test. Red boxes represent participants with a higher adherence to the MD and blue boxes low adherence.

**Figure 6 nutrients-13-00636-f006:**
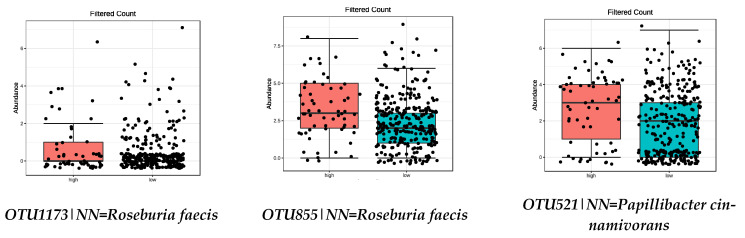
Bacterial species that were significantly more abundant in the group with high NUTS intake (FDR < 0.05) by metagenomeSeq test. Red boxes represent participants with a higher adherence to the MD and blue boxes low adherence.

**Table 1 nutrients-13-00636-t001:** Baseline characteristics of the population separated by adherence to the MD (Mediterranean diet).

Variables	High Adherence (3rd Tertile) (*n* = 94)	Low Adherence (1st Tertile) (*n* = 128)	*p* Value
Age (y)	47.3 ± 1.2	41.7 ± 0.9	0.020 *
BMI	27.7 ± 0.6	29.9 ± 0.6	0.002 *
Glucose (mg/dL)	95 ± 2	93 ± 1	0.394
Total cholesterol (mg/dL)	209 ± 4	208 ± 4	0.822
HDL (mg/dL)	59 ± 1	57 ± 1	0.261
LDL (mg/dL)	133 ± 4	132 ± 3	0.904
Triglycerides (mg/dL)	87 ± 5	93 ± 4	0.336
HOMA-IR	1.4 ± 0.15	1.6 ± 0.1	0.011 *
Carbohydrate intake (%)	54.5 ± 0.7	54.7 ± 0.6	0.135
Protein intake (%)	21.8 ± 0.4	22.2 ± 0.4	0.286
Fat intake (%)	23.6 ± 0.5	23.0 ± 0.4	0.085
Fiber intake (g/day)	35.9 ± 1.6	24.5 ± 0.7	0.000 *
Energy intake (kcal/day)	2932 ± 106	2872 ± 84	0.680

The *p* value was calculated depending on the distribution of the variables by T-Student or Mann–Whitney test. * The score is significant at the 0.05 level.

**Table 2 nutrients-13-00636-t002:** Bacterial species with a significant relation with adherence to the MD (FDR (False discovery rate) < 0.05) by metagenomeSeq test.

High Adherence (3rd Tertile)		Low Adherence (1st Tertile)	
SPECIES	FDR	SPECIES	FDR
*Bifidobacterium animalis*	1.21 × 10^−7^	*OTU100|NN = Eubacterium saphenum GU427005|D = 91*	4.44 × 10^−5^
*Bacteroides cellulosilyticus*	4.47 × 10^−7^	*OTU375|NN = Succinivibrio dextrinosolvens Y17600|D = 97*	0.0001
*OTU946|NN = Paraprevotella clara AB331896|D = 86.8*	1.72 × 10^−5^	*OTU759|NN = Gordonibacter pamelaeae AB566419|D = 87.6*	0.0005
*OTU1682|NN = Oscillibacter valericigenes AB238598|D = 91.1*	3.42 × 10^−5^	*OTU11|NN = Butyricicoccus pullicaecorum EU410376|D = 89*	0.0002
*OTU1065|NN = Oscillospira (Flavonifractor) plautii Y18187|D = 86.6*	3.42 × 10^−5^	*Christensenella minuta*	0.0020
*OTU1173|NN = Roseburia faecis AY804149|D = 94.9*	0.0008	*Parabacteroides goldsteinii*	0.0073
*OTU1517|NN = Catabacter hongkongensis AB671763|D = 87*	0.0008	*OTU1625|NN = Anaerotruncus colihominis DQ002932|D = 89*	0.0120
*OTU1296|NN = Ruminococcus bromii DQ882649|D = 92.3*	0.0120	*Alistipes timonensis*	0.0155
*Erysipelatoclostridium ramosum*	0.0176	*Prevotella corporis*	0.0192
*OTU521|NN = Papillibacter cinnamivorans AF167711|D = 89*	0.0463		

**Table 3 nutrients-13-00636-t003:** Bacterial species and OTUs (Operational Taxonomic Units) significantly related (FDR < 0.05) with the main food groups of the MD, by comparing high and low tertiles of the intake of each food group by metagenomeSeq analysis.

SPECIES	FDR
**High intake of FIBRE**	
*OTU1026|NN = Catabacter hongkongensis AB671763|D = 82.4*	8.48 × 10^−11^
*OTU1682|NN = Oscillibacter valericigenes AB238598|D = 91.1*	6.73 × 10^−5^
*OTU1173|NN = Roseburia faecis AY804149|D = 94.9*	8.38 × 10^−5^
*OTU1065|NN = Oscillospira (Flavonifractor) plautii Y18187|D = 86.6*	0.0018
*Oscillibacter valericigenes*	0.0045
**High intake of LEGUMES**	
*OTU222|NN = Ruminococcus bromii DQ882649|D = 89.9*	7.45 × 10^−5^
*OTU1712|NN = Catabacter hongkongensis AB671763|D = 84*	0.0011
*Bacteroides cellulosilyticus*	0.0001
**High intake of VEGETABLES**	
*OTU1517|NN = Catabacter hongkongensis AB671763|D = 87*	1.19 × 10^−11^
*OTU1026|NN = Catabacter hongkongensis AB671763|D = 82.4*	1.43 × 10^−6^
*OTU521|NN = Papillibacter cinnamivorans AF167711|D = 89*	0.0019
**High intake of FRUIT**	
*OTU1517|NN = Catabacter hongkongensis AB671763|D = 87*	0.0008
*OTU1173|NN = Roseburia faecis AY804149|D = 94.9*	0.0012
**High intake of NUTS**	
*OTU1173|NN = Roseburia faecis AY804149|D = 94.9*	8.11 × 10^−5^
*OTU855|NN = Roseburia faecis AY804149|D = 95.5*	0.0001
*OTU521|NN = Papillibacter cinnamivorans AF167711|D = 89*	0.0024

**Table 4 nutrients-13-00636-t004:** Metabolic activity differences between the tertiles with lower and higher adherence to MD by Tax4Fun test.

K0 Name	Definition	Log2FC	*p* Value	FDR
K00720	Ceramide glucosyltransferase	1.803	2.93 × 10^−7^	0.001
K01884	Cysteinyl-tRNA synthetase	1.792	8.67 × 10^−7^	0.002
K03653	N-glycosylase/DNA lyase	1.6569	1.57 × 10^−6^	0.002
K02106	Short-chain fatty acids transporter	1.0933	2.26 × 10^−5^	0.03
K07486	Transposase	0.9605	4.51 × 10^−5^	0.04

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
