# Peer review of "Gut Microbiota Bacterial Species Associated with Mediterranean Diet-Related Food Groups in a Northern Spanish Population"

_nutrients, 2021, doi:10.3390/nu13020636_

Round 1

Reviewer 1 Report

The manuscript submitted by Rosés et al., titled: “Bacterial species associated with Mediterranean diet -related food groups in a Spanish population” is aiming at studying the relationship between intake of food groups characteristic of the Mediterranean diet and the gut microbiome profile in Spanish individuals. The reviewer would like to raise the following points in regards to the aforementioned manuscript submitted:

  1. The title is not accurately reflecting the aim of the study. There is no reference to the gut microbiome per se and thus the way the title is phrased could refer to bacteria that are found in the food groups. The title is conceptually written in poor English and needs revision to reflect what the study aimed to do and has to report.
  2. The authors state several times (in the abstract and the opening sentence of the introduction) that there is a direct effect of the gut microbiome on the health of the host. This is scientifically inaccurate. The microbiome indeed is associated with increased or reduced risk of disease and as such there are more desirable or less desirable microbiome profiles. The manner and mechanism through which these effects are extended involve metabolites and in turn metabolic responses and flexibility, signaling and the immune system. So, there is an indirect in actuality effect on the microbiome on health.

2.a. In the introduction the phrase “interactive associations” needs clarification.

  1. The language and the style of the manuscript must be improved there are often simplifications with the use of colloquial and incorrect language. For example, the authors refer to “Western diseases”. While the term “Western diets” is an accepted one from a scientific perspective this is not the case with “Western diseases”. Actually, there are more people for example with T2DM in India and China combined than any other place on the globe.

Grammar and syntax also need improvement.

3.a. The word “participant” should replace the word subject.

  1. Line 77: the work does not relate A to B, but rather investigates the relationship between A and B. When research work is undertaken even though we may have a preconceived notion on the potential outcome based on biological plausibility and existing literature, we still are researching thus we cannot state in advance that there is a relationship between dependent and independent variable.
  2. BMI is an index and is dimensionless, that is why we calculate BMI and we do not measure it. As a result, there are no units for BMI. Kg/m2 is a force unit and certainly we do not measure body surface when producing BMI values as the m2 implies.
  3. Inclusion criteria of the population should also be reported. Furthermore, addressing confounding factors such as smoking status, medication, supplement(s) intake especially when performing dietary intake analysis are all important. Furthermore, how did the authors address the span in their cohort between males and females in terms of the unbalanced sample size?
  4. Mere reference on the characteristics and demography of the participating cohort to a previously referenced report is not sufficient. The reader needs to briefly obtain direct information on the characteristics and demography of the participating population preferably in a tabulated format.
  5. Discuss the limitations of the study in terms of its design. For instance, what was the mode of selection for the participants? To what extent is this population in this locale in Spain representative? If it is not, then how and to what extent is it different? What are the specific characteristics that may render it on the more unique side? What was the socioeconomic background on the participants and was that considered along with any variations in the authors’ analyses?
  6. The authors mentioned that they used the Obekit cohort (normal weight, overweight and obese subjects) in this work that they report. Apparently, there is significant differences as per BMI and potentially body composition. These are in their own right significant confounding factors that independently are associated with altered microbiome profiles. If there is no statistical analysis employed to tease apart or correct and normalize the effects of BMI and obesity, then the results are undermined and the conclusion potentially misleading.
  7. How did the authors control for/address participant misreporting on the food intake data?
  8. From the results section it becomes clear that there is a fibre effect on the microbiota profile of the participants. This is not necessarily unique to the Mediterranean Diet, however. Other plant-based diets can be very rich in fibre.
  9. Furthermore, there is significant evidence in the literature indicating that high fibre diets produce a more favorable microbiome profile in the gut thus the finding in discussion is neither surprising nor novel.

Author Response

The manuscript has been reviewed by a native English speaker

Reviewer 2 Report

The manuscript present an interesting study. But I would recommend authors to consider the following general and specific remarks to improve the quality of the manuscript:

Food consumption patterns in Spain and in most regions and countries have changed markedly in the last years, and differ somewhat at present from the dietary habits influenced by gastronomic culture, and even from traditional and healthy Mediterranean Diet. Therefore, would have been interesting to adress the differences between dietary habits, influenced by gastronomic culture and habits of Spanish region and Mediterranean Diet.

Please take into consideration that:

Erysipelotrichi is a class within the Firmicutes. Clostridium ramosum is a member of the Erysipelotrichi. It has been proposed to assign C. ramosum and four related species to the new genus Erysipelatoclostridium. The microorganism was renamed Erysipelatoclostridium ramosum in 2013.

Author Response

(The authors gave the same response as above.)

Round 2

Reviewer 1 Report

Minor mistakes are identified in the revised version of the paper and included below. To ensure a higher quality of the paper, the reviewer would recommend a more careful proofreading of the text by the authors and a native English speaker.

Ln 58: Non-communicable diseases are related to, but not derived from Western diets. Other factors such as genetics, sedentary lifestyle, smoking etc. are involved in the development of NCDs.

Ln 423: “which may be a little bias” Language could be improved, for example it could read “potentially introducing bias”

Ln 427: Should be corrected to read “be considered”

Ln 71: The reference should be “Mitsou et al.
